# Flow Cytometry as a New Accessible Method to Evaluate Diagnostic Osmotic Changes in Patients with Red Blood Cell Membrane Defects

**DOI:** 10.3390/biomedicines12071607

**Published:** 2024-07-19

**Authors:** Asunción Beltrán, María Sánchez-Villalobos, Eduardo Salido, Carmen Algueró, Eulalia Campos, Ana Belén Pérez-Oliva, Miguel Blanquer, José M. Moraleda

**Affiliations:** 1Instituto Murciano de Investigación Biosanitaria (IMIB)—Arrixaca, 30120 Murcia, Spain; lalycampos19@gmail.com (E.C.); anab.perez@imib.es (A.B.P.-O.); jmoraled@um.es (J.M.M.); 2Servicio de Hematología, Hospital Clínico Universitario Virgen de la Arrixaca, 30120 Murcia, Spain; eduardoj.salido@carm.es (E.S.); calgueromartin@gmail.com (C.A.); miguelblanquer@um.es (M.B.); 3Hematología, Trasplante Hematopoyético y Terapia Celular, Departamento de Medicina, Universidad de Murcia, 30001 Murcia, Spain

**Keywords:** congenital hemolytic anemia, red blood cell membrane, hereditary spherocytosis

## Abstract

Hereditary spherocytosis (HS) is a membranopathy that impacts the vertical junctions between the cytoskeleton and the plasma membrane of erythrocytes. The gold standard method for diagnosing it is osmotic gradient ektacytometry (OGE). However, access to this technique is scarce. We have devised a straightforward approach utilizing flow cytometry to quantify variations in an osmotic gradient, relying on FSC-H/SSC-H patterns. We studied 14 patients (9 pediatric, 5 adults) and 54 healthy controls (16 pediatric, 38 adults). After assessing the behavior of the samples in several osmolar gradients we selected for the study the 176, 308, and 458 mOsm/kg levels as hypo-osmolar, iso-osmolar, and hyper-osmolar references. We then selected the iso-osmolar point for assessment to determine its efficacy in discriminating between patient and control groups using a receiver operating characteristic curve. In the pediatric group, the area under the curve (AUC) was 1.0, indicating 100% sensitivity and 93.3% specificity. Conversely, in the adult group, the AUC was 0.98, with 80% sensitivity and 90.9% specificity. We introduce a method that is easily replicable and demonstrates high sensitivity and specificity. This technique could prove valuable in the diagnosis of spherocytosis.

## 1. Introduction

The red blood cell (RBC) is a specialized cell characterized by a certain degree of deformability, enabling it to navigate through narrow microcapillaries. This physical property is associated with the integrity of the plasma membrane and the proteins comprising the cytoskeleton [1].

Membranopathies encompass a group of congenital hemolytic anemias that impact the proteins constituting the cell membrane and the cytoskeleton [2]. Additionally, these conditions involve dysfunction in certain membrane channels and transporters responsible for regulating cellular homeostasis [3]. They are characterized by being a heterogeneous group of clinical entities with presentations ranging from a trait to severe clinical forms [3].

The most prevalent disorder among the membranopathies is hereditary spherocytosis (HS), with a prevalence rate of 1 in 2000 births in the Caucasian population [4]. This pathology impacts proteins associated with the vertical junctions of the membrane and influences the genes *ANK1, SPTB, SPTA1, EPB4*, and *SCLA41* [5]. Inheritance usually follows a dominant pattern (75%), while 25% of cases are attributed to recessive or de novo inheritance mechanisms [3,5].

The diagnosis is typically established through an assessment of the patient’s family history, clinical presentation, and abnormalities detected in specific laboratory examinations. The age at which the diagnosis is made may occur during childhood or adulthood, with variations in clinical manifestations being noted [4,6]. Among the specific tests used for diagnosing erythrocyte membrane protein disorders, the Eosin-5-maleimide (EMA) binding test stands out as a highly specific and sensitive diagnostic tool. The interaction involves the fluorochrome EMA binding to the K430 residue of Band 3, as well as binding to the proteins CD47 and RhAG [7,8].

Among the specific diagnostic tests for HS and other membranopathies, the osmotic gradient by ektacytometry (OGE) is noteworthy [5,9]. This technique can assess the deformability of red blood cells by analyzing specific light diffraction patterns and calculating indices that aid in diagnosing a wide range of pathologies [9,10,11,12,13,14]. This method is regarded as a gold standard; however, its utilization is challenging due to the high expenses involved and the scarcity of qualified personnel with expertise in this area.

Flow cytometry is a commonly employed laboratory technique that enables the rapid analysis of multiple cellular populations per second. The biconcave shape of the RBC results in a bimodal distribution in the forward scatter parameter (FSC-H), which serves as an indicator of cell size [15,16,17,18]. The variability in the distribution of this parameter has enabled certain authors to characterize the shape of RBC cells under various osmolar conditions [19,20], as well as in individuals with kidney disease, sepsis, and diabetes [17]. The examination of empirical models employing a dual orthogonal laser in the flow cytometer has facilitated the characterization of the deformation of RBC within the cytometer and the quantification of the distribution of the FSC-H pattern [18].

In this study, an analysis was conducted on the FSC-H/SSC-H parameters under different osmotic conditions both in healthy controls and patients. Our observations revealed notable differences that suggest the potential of utilizing this technique as a diagnostic tool for HS in the future.

## 2. Materials and Methods

### 2.1. Patients

Nine pediatric and five adult patients diagnosed with HS based on family history, clinical symptoms, laboratory findings, and an EMA or Osmotic fragility positive test were examined at the Erythropathology Unit of the University Hospital Virgen de la Arrixaca (Murcia, Spain) (Figure 1).

### 2.2. Control Group

Fifty-four blood samples from healthy age-paired controls (HC) were evaluated, comprising 38 adults and 16 children.

This study received approval from the Research and Ethics Committee of Hospital Virgen de la Arrixaca and was carried out in accordance with the principles outlined in the Declaration of Helsinki. All participants provided their informed consent for the collection of blood samples and clinical and laboratory data, which were subsequently anonymized.

### 2.3. Work-Flow

#### 2.3.1. RBC Indices, Reticulocyte Count and Biochemical Analysis

The clinical diagnostic features and follow up of the patients were obtained from the primary anemia registry of our laboratory.

Biological Samples were analyzed using a hematological analyzer (Sysmex XN20, Kobe, Japan) evaluating the following RBC indices: hemoglobin (Hb), hematocrit, mean cell volume (MCV), mean corpuscular hemoglobin concentration (MCHC), mean hemoglobin concentration (MHC), red cell distribution width (RDW), absolute reticulocyte count, and percentage of reticulocytes.

Morphological analysis was carried out by blood smears stained with May Grümwald Giemsa (MGG) and examined by a hematologist, less than 24 h after sample collection.

Biochemistry studies included serum levels of bilirubin (total, indirect and direct), lactate dehydrogenase (LDH), iron, ferritin, and soluble transferrin receptor. Direct agglutination test (DAT), hemoglobin electrophoresis, and/or G6PD enzyme activity measurements were performed on the patients for whom it felt relevant for their diagnosis.

#### 2.3.2. Eosin-5′-maleinide (EMA) Dye Test by Flow Cytometry

To perform the EMA test 1 µL of EDTA anticoagulated blood was suspended in 200 µL of phosphate-buffered saline (PBS). Twenty-five µL of Eosin 5′-Maleimide (EMA) solution (Sigma Aldrich, Darmstadt, Germany) were added from the working solution (0.5) mg/mL) in DMSO (WAK-Chemie Medical GmnH, Steinbach, Germany), and it was incubated for 1 h in the dark at room temperature (rt). Subsequently, 2 mL of PBS was added, and it was centrifuged for 5 min at 1650 g rt. The supernatant was removed and suspended in 300 mL of PBS. The sample was measured using a cell analyzer, cytometer BD LSR Fortessa X-20 (Becton Dickinson, East Rutherford, NJ, USA). The light scattering parameters FSC/SSC were measured on a logarithmic scale, using a selection gate in the RBC region, and the mean fluorescence intensity (MFI) was determined using the FL-1 fluorescence channel. A total of 10,000 events per region were acquired from each tube. Analysis was performed using FACSDiva software version 6.0 (BD Biosciences, San Jose, CA, USA). The result obtained is the ratio between the mean MFI of the controls and that of the patient. The study samples were compared with 5 healthy controls matched with the patient age. The cut-off values for the test to be considered positive were those in which the mean fluorescence ratio was above 21%. Values between 16 and 21% were considered intermediate, and values up to 16% were considered normal or negative [7].

#### 2.3.3. Osmolar Curve by Flow Cytometry

##### Samples

Blood was collected in a 2.7 mL tube coated with K+EDTA by venipuncture and analyzed within 24 h after collection. All samples were diluted according to the methodology outlined by Won et al., which includes creating a cell suspension through a two-step dilution process [21]. In the first step, the volume in µL was taken as determined by the following formula:Blood volume (µL) = 130/(Number of RBC µL × 10^6^)
and suspended in 1000 µL of saline solution (308 mOsm/kg). Subsequently, a volume of 10 µL of the suspension was introduced into a tube containing 400 µL of saline solution and analyzed by flow cytometry (BD LSR Fortessa X-20, Becton Dickinson). An acquisition time of 250 s was set up. FSC-H/SSC-H parameters were adjusted on a logarithmic scale, while the linear parameters remained active. An acquisition gate was set in the region of red blood cells (RBC). A low flow rate of 12 µL/min was employed. A kinetic curve was conducted with data acquisition intervals of 30 s. During each interval, 100 µL of deionized distilled water was introduced into a hypo-osmolar medium, resulting in a gradual decrease in osmolarity at the following points: 308, 246, 205, 176, 136, 123, and 112 mOsm/Kg. A 0.47 M NaCl solution was incrementally added to the hyper-osmolar curve, resulting in an increase in osmolarity at the following points: 308, 349, 415, 458, 473, 578, 585, and 592 mOsm/kg. Osmolarity was checked with an osmometer.

The flow cytometry data analysis was conducted utilizing Flow Jo 10.6 software (Treestar Inc., Ashland, OR, USA). The region corresponding to erythrocytes was defined using a dot plot of forward scatter (FSC-A) versus side scatter (SSC-A) on a logarithmic scale. Subsequently, the regions associated with each data point on the kinetic curve, categorized as either hypo-osmolar or hyper-osmolar, were plotted using an FSC-H linear/time in seconds histogram. The mean fluorescence intensity (MFI) was computed for FSC-H and SSC-H parameters.

#### 2.3.4. Statical Analyses

The study involves a descriptive analysis encompassing the mean, standard deviation, minimum and maximum values, ranges, and receiver operating characteristic (ROC) curve. The normality of variables was evaluated using the Kolmogorov–Smirnov normality tests for both healthy control subjects and patients at each data point along the curve. Statistical comparisons between each data point on the curve of the healthy controls and patients were conducted utilizing the Kruskal–Wallis test. A *p*-value < 0.05 was deemed to be statistically significant. A comparison of curve points at 308, 176, and 458 mOsm/kg between healthy controls and patients was conducted utilizing the unpaired non-parametric Mann–Whitney U test. A *p*-value < 0.05 was deemed statistically significant. The analyses were conducted utilizing IBM-SPSS Statistics v29.0 (IBM Corp., Armonk, NY, USA) and GraphPad Prism v9.0 (GraphPad Software, San Diego, CA, USA).

## 3. Results

### 3.1. Patients

Table 1 shows the analytic characteristics of the RBC at diagnosis of the patients according to their splenectomy status. The four splenectomized patients displayed mean RBC levels and reticulocyte count within the normal range. In contrast, ten patients with HS without splenectomy exhibited moderate to severe anemia with an elevated reticulocyte count and compensated hemolytic anemia.

### 3.2. Eosin-5′-maleinide (EMA) Dye Test by Flow Cytometry

EMA results from 12 patients in which the test was performed are shown in Table 2. The remaining two patients were diagnosed in their childhood by the clinical feature and osmotic fragility test.

All patients were positive (≥21% decrease).

### 3.3. Osmolar Curve by Flow Cytometry

A new methodology to measure the changes that RBCs undergo at different osmolarities under the constant pressure exerted by the hydrodynamic forces of the cytometer flow system was devised. The shear forces in the flow chamber of the cytometer change the resting positions of the RBCs causing them to assume an elliptical shape without energy loss, a laser beam interrogates the cells and FSC-H and SSC-H photodetectors are activated and measure the intensity of the generated pulse. To measure cytometer changes in RBCs in a hypo-osmolar, hyper-osmolar, and iso-osmolar media, we performed a kinetic curve. We consider that the FSC parameter represents the major axis of the elliptical shape adopted and the SSC parameter represents the minor axis. The ratio of these two parameters will allow for the calculation of the elongation index by flow cytometry [EI-FC = (FSC-H − SSC-H)/(FSC-H + SSC-H)] as is shown in Figure 2. The ability of flow cytometry to measure the shape of the RBC by their FSC/SSC parameters can be somewhat compared to the ektacytometry measurements. And so, we stipulated that modification of the ektacytometry elongation index considering those flow cytometry parameters could be feasible and offer a more widely adaptable technique to study RBC deformability that could be easily included in the diagnostic streamline of the primary anemias.

#### 3.3.1. Osmotic Changes in RBC in Healthy Controls

The elongation index was calculated by utilizing the FSC/SSC parameters at different positions along the osmotic gradient, following the methodology outlined in Figure 2. The osmotic gradient of the adult and pediatric control groups, presenting the average relationship and standard deviation at each data point, is depicted in Figure 3. It is noteworthy that this variation escalates as the distance from the iso-osmolar point increases. In the adult control group, there is an observed increase in deviation at the extremities of the hypo-osmolar curve (Figure 3A). In the pediatric control group, higher deviations are observed at points 205 and 176 mOsm/Kg (Figure 3B).

We then checked whether the observed differences had statistical significance. In adults, comparing with the EI-FC ratio at the iso-osmolar point (308 mOsm/kg), significative differences were found in the hypo-osmolar points ≤ 176 mOsm/kg, left bars (Figure 4A) and in the hyperosmolar ones ≥ 458 mOsm/kg right bars (Figure 4A). In children, significative differences were found in the ≤136 mOsm/kg left bars (Figure 4B) and ≥473 mOsm/kg point right bars (Figure 4B). These results were reassuring in terms of the capacity of the technique to identify the RBC morphological changes under different osmotic conditions.

#### 3.3.2. Compare Osmotic Changes between HS Patients and Control Group

A comparative analysis was performed to examine the mean differences between healthy controls and patients both adult and pediatric, at three distinct points on the curves where the differences from the iso-osmolar point become increasingly apparent: 176, 308, and 458 mOsm/kg (Figure 4). The iso-osmolar point ratio exhibited a significant decrease in both adult and pediatric patients in comparison to the control group (*p* < 0.0001), both in adult (Figure 5A) and pediatric (Figure 5B) patients. At 176 mOsm/kg, there was a statistically significant difference (*p* < 0.05) in the adult group and (*p* < 0.00001) in the pediatric group compared to the control group (Figure 5). At an osmolality level of 458 mOsm/kg, significant differences were also observed between patients and the control group (*p* < 0.05) in both adult and pediatric populations.

We proceeded to compare our diagnostic test’s accuracy with that of a reference test. A ROC curve was constructed using data points measured at 176, 308, and 458 mOsm/kg and was compared with the EMA technique in the patient and control cohorts. The results of the EMA assessment were recorded as either positive or negative. In two adult patients, the diagnosis of hereditary spherocytosis (HS) was established through a positive osmotic fragility test result and their medical history, which included a prior diagnosis of HS in childhood. To ascertain the optimal cut-off point the area under the ROC curve was evaluated as a measure of the overall performance of the diagnostic test.

For the pediatric population, the osmolality value of 176 mOsm/kg showed an AUC of 0.9762, indicating a sensitivity of 100% and a specificity of 66.7%. At the iso-osmolar point, the AUC was 1.0, indicating a sensitivity of 100% and a specificity of 93.3%. At the threshold of 458 mOsm/kg, a decrease in both sensitivity (57%) and specificity (88.89%) of the technique is noted, leading to an observed AUC of 0.82 (Figure 6).

In the case of adults, the 176 mOsm/kg point demonstrated an AUC of 0.8364, with a sensitivity of 18% and a specificity of 80%. At the iso-osmolar point, the AUC was 0.98, demonstrating a sensitivity of 80% and a specificity of 90.91%. At the point 458 mOsm/kg, the sensitivity decreases to levels approaching 0, while the specificity of the technique was 80%, with an AUC of 0.74 (refer to Figure 6).

The iso-osmolar point provides enhanced sensitivity and specificity, particularly in the pediatric population. In adults, the values associated with the iso-osmolar point may be deemed acceptable for diagnostic testing.

In the patient cohort, both pediatric and adult individuals exhibited values below the threshold established in the ROC curve at the iso-osmolar point (0.098 for children and 0.1035 for adults). To visually analyze the osmotic gradient, a comparison was made between the mean values of the adult and pediatric control groups and those of the patients. The curves depicting pediatric patients with (HS) exhibited a decrease in the AUC, as demonstrated in Figure 7. In adult patients, a reduction in the AUC was also noted in comparison to the control group, in one adult patient we observed in hyperosmolar points an increase in the EI-FC ratio over the control group.

A ratio was computed to compare the iso-osmolar point ratio of the patient under consideration with the mean ratio of the control group at the same point. The outcomes are expressed as a percentage reduction in EI-FC, calculated using the following formula:Ratio EI-FC = EI-FC patient/EI-FC controls(1)

A decrease in EI-FC was observed in all patients. This reduction was particularly pronounced in the pediatric patient cohort, exhibiting a more substantial decrease in proportion compared to the adult patients (Table 3).

We then investigated the potential relationship between the calculated EI-FC ratio in patients with HS and the RBC parameters from the same sample and found no significant correlation. However, we confirmed the already described good predictive value of the reticulocyte count and RDW in HS patients (Figure 8).

## 4. Discussion

In this study, we compared the size/complexity values by flow cytometry (FC) in RBCs at different osmolarities between a healthy population of children and adults and a population of HS patients.

Ektacytometry, which measures diffraction patterns of light through laser beam on a suspension of RBCsat constant shear pressure, provides four fundamental parameters: EImax (maximum rate of elongation at isotonic osmolality); Omin (hypotonic osmolality where EI is minimum, reflecting the surface area/volume ratio); Ohyper (osmolality value of the hypertonic region corresponding to 50% of EImax, related to cell hemoglobin concentration (CHCM)), and AUC (area under the curve, defined between Omin and the osmolality point at 500 mOsm/kg) [13,22]. This technique has allowed the development of curves for different congenital hemolytic anemias [5,9,12,13,23,24,25]. However, trained personnel is needed and equipment availability is scarce.

The bimodal distribution observed in the forward scatter histogram (FSH-H) of RBCs is associated with their biconcave shape. Several authors have investigated this phenomenon to determine the morphology of RBCs under various circumstances, including in individuals with sepsis. [17] Other investigators have employed cytometry to investigate the behavior of RBC in different osmolar solutions, aiming to observe alterations in RBC morphology [20]. Gienger and colleagues showed that the bimodal shape of RBC in the FSC-H histogram is related to the hydrodynamic forces that allow the cell to travel in the flow and that the sheer forces exerted on the cell allow it to acquire an elliptical shape, where its deformability can be measured [18]. Imaging cytometry has also been used to view the shape of RBC [26].

In our study, significant changes were observed in the FSC/SSC parameters of RBCs from healthy individuals as the osmolarity of the surrounding medium varied, creating a gradient. Variability at specific points in this gradient was noted, likely due to the inherent inter-variability of RBCs, including both young and old cells [24]. This variability may also be linked to cellular lysis at certain gradient points, causing deviations from a normal distribution.

We examined and compared three points within the gradient (176, 308, and 458 mOsm/kg) with patient data from both adult and pediatric populations, finding significant differences at all three points between the control and patient groups. These points were analyzed using a ROC curve to measure specificity and sensitivity [27]. At the iso-osmolar point, high sensitivity and specificity were observed in both pediatric and adult cohorts (100% and 93.3% vs. 80% and 90.1%). The variability in sensitivity and specificity could be attributed to the 176 mOsm/kg point, influenced by the variability seen in both controls and patients within the distribution (see Appendix A) due to hemolysis—a phenomenon described at this osmolarity on the OGE curve, where an increase in the coefficient of variation at the Omin with a variability of 5–8% in children was observed [9]. The iso-osmolar point showed reduced variability and consistent performance in both pediatric and adult populations, making it a promising diagnostic marker.

The values at 308 mOsm/kg (EI-FC) were correlated with red blood cell indices from the same blood sample, but no strong or statistically significant correlation was found, likely due to the functional nature of the test.

The iso-osmolar point allows for sensitive and specific detection of differences between control and patient populations. However, standardizing the technique would require a large sample size due to the varied clinical presentations and the need for studies using OGE to ensure optimal sensitivity and specificity.

There is an increasing demand for novel, uncomplicated, and readily available diagnostic methods. The new test allows us to demonstrate differences between normal and abnormal RBC populations and permits sensitive and specific discrimination, making it useful for the diagnosis of HS. In addition to molecular diagnosis, these methodologies could assist in elucidating particular issues arising from the significant heterogeneity of clinical manifestations.

## Figures and Tables

**Figure 1 biomedicines-12-01607-f001:**
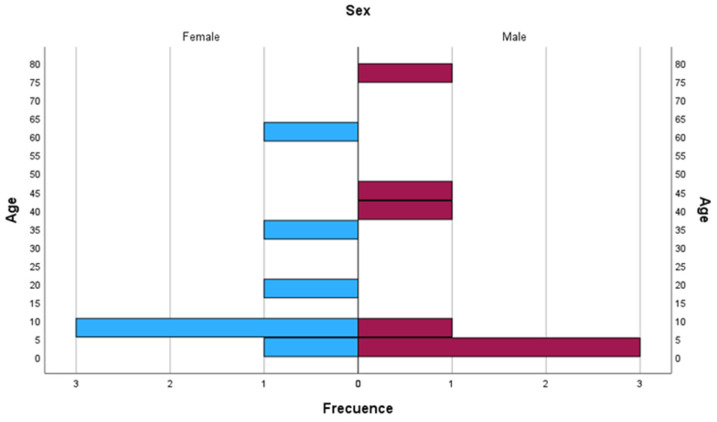
Distribution of age and gender among patients diagnosed with hereditary spherocytosis (HS).

**Figure 2 biomedicines-12-01607-f002:**
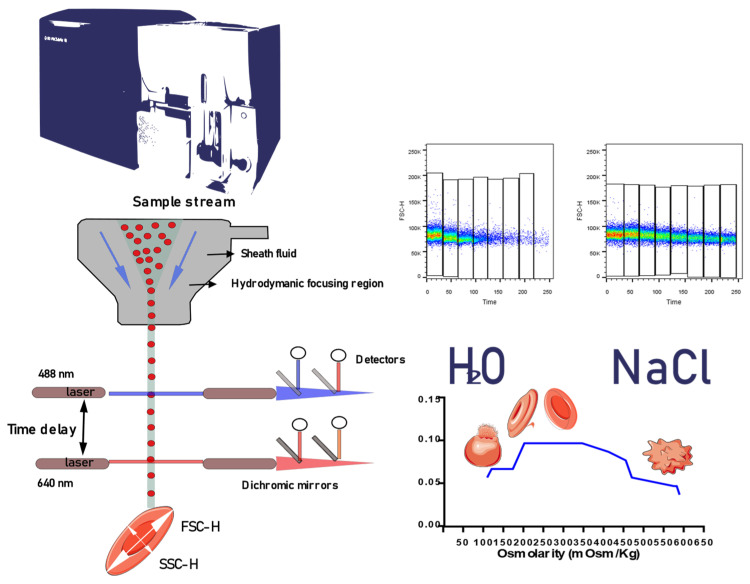
The technique operates on a simple principle: a laser beam is directed onto red blood cells within a flow cell. As the cells are subjected to hydrodynamic forces within the flow system, they transition from a resting position to an elliptical shape due to shear force. By analyzing patterns FSC/SSC, we can observe changes in these patterns under constant shear force. The ratio of these two parameters yields the elongation index calculated by flow cytometry (EI-FC = (FSC-H − SSC-H)/(FSC-H + SSC-H)). Graphically, the Y-axis represents the EI-FC, while the X-axis denotes various osmolarities.

**Figure 3 biomedicines-12-01607-f003:**
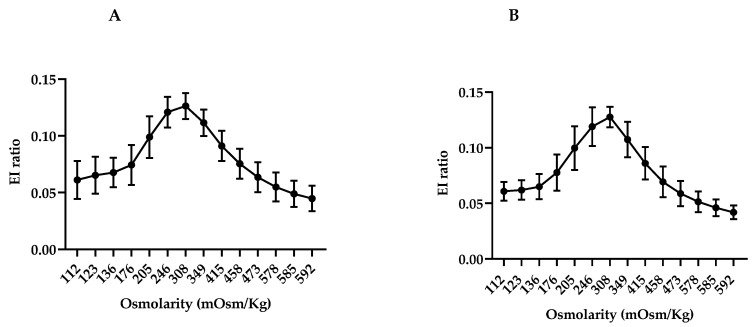
The elongation index by flow cytometry (EI-FC) was assessed in control cohorts comprising healthy adults and children. Peripheral blood samples obtained from a cohort of healthy adults (HA) (*n* = 38) and healthy children (HC) (*n* = 16) underwent flow cytometry analysis to assess the FSC/SSC parameters. The Y-axis depicted the EI-FC, whereas the X-axis represented different osmolarities. (**A**) HA curve of the adults. (**B**) HC curve of the children.

**Figure 4 biomedicines-12-01607-f004:**
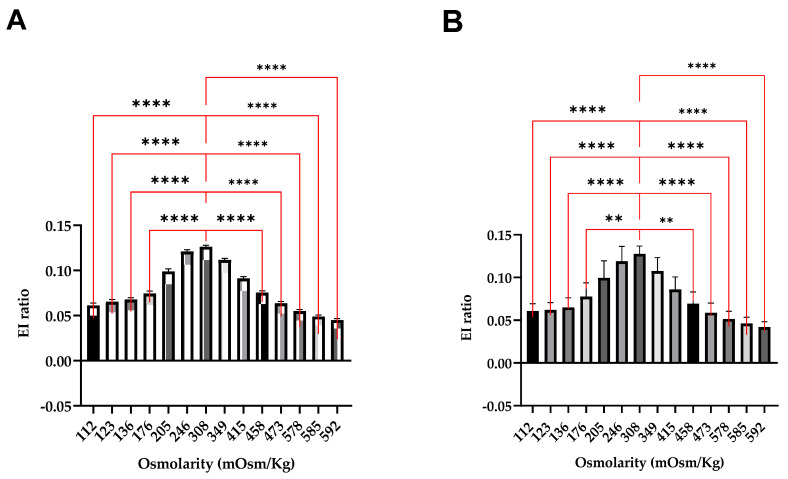
Comparison of gradient points with the iso-osmolar point in adult and pediatric control populations (**A**,**B**). Each bar represents the cumulative mean of the curve points. Unpaired Kruskal–Wallis tests were applied as needed. ** *p* < 0.01 and **** *p* < 0.00001.

**Figure 5 biomedicines-12-01607-f005:**
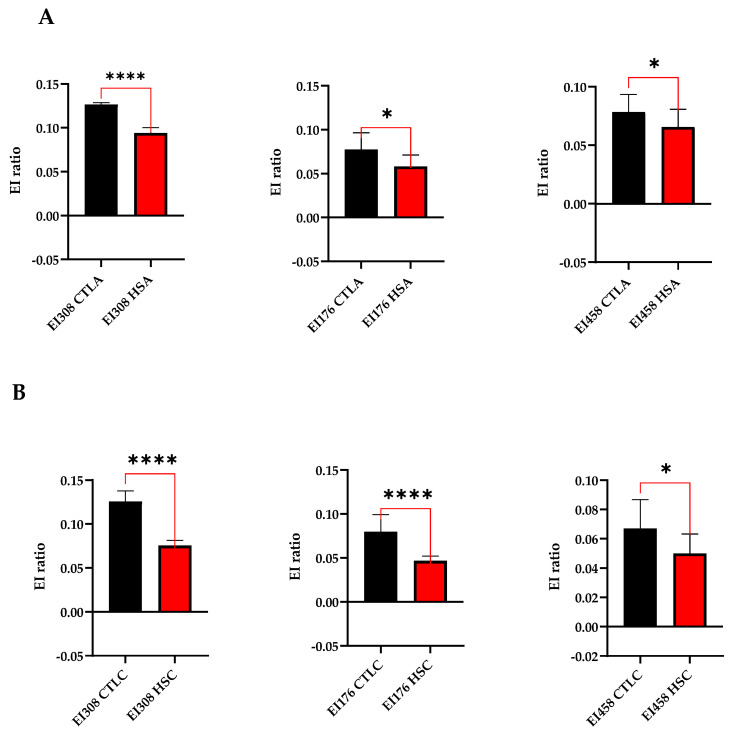
(**A**) Adult population and (**B**) pediatric population comparison of means was conducted between the control group (CTL) and the patient group (HS) staging at the osmolality levels of 176, 308, and 458 mOsm/kg. Unpaired Mann–Whitney tests were conducted where applicable (* *p* < 0.05 and **** *p* < 0.0001).

**Figure 6 biomedicines-12-01607-f006:**
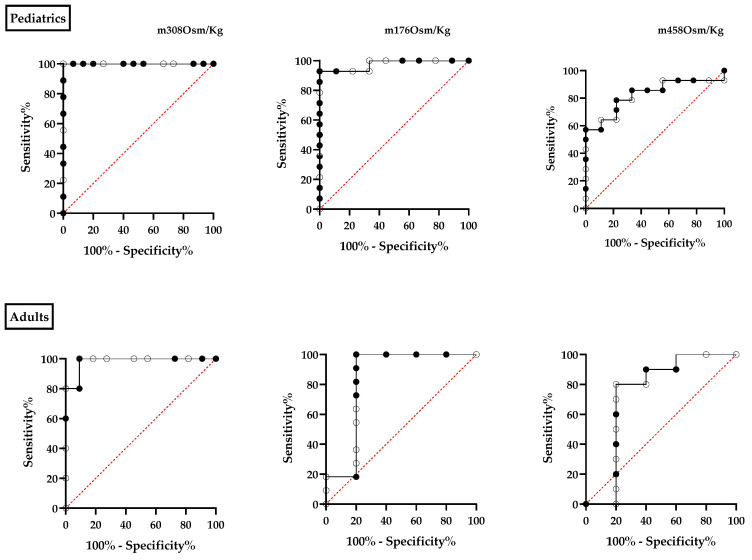
Displays the ROC curve for the data points at 176, 308, and 458 mOsm/Kg. The values obtained from the EI-FC of both patients and controls were compared with the positive and negative results of the EMA binding test in the respective groups.

**Figure 7 biomedicines-12-01607-f007:**
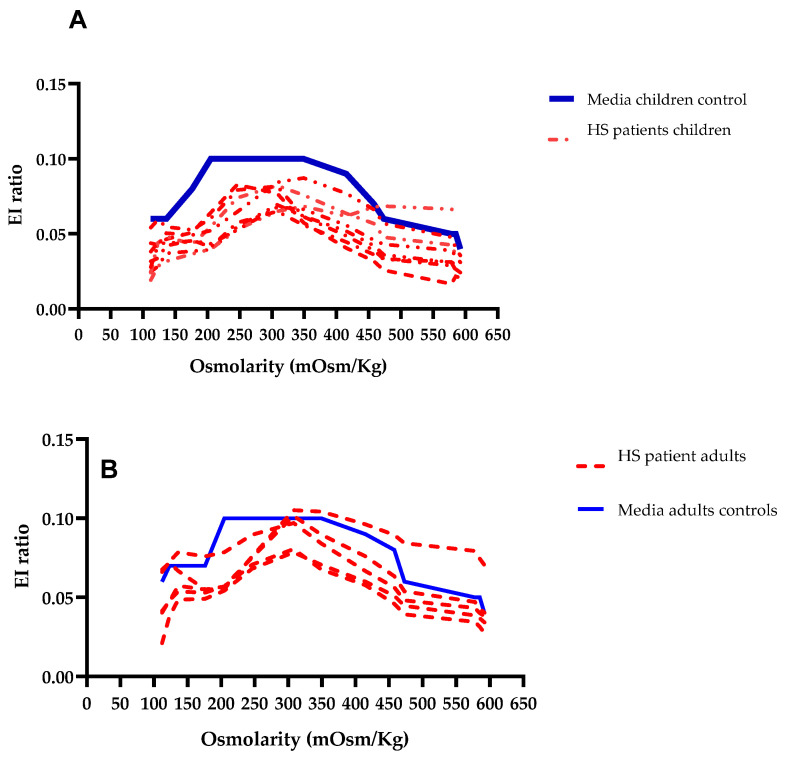
Illustrates the osmotic gradients derived from the average values of the adult and pediatric control groups in comparison to the curves observed in patients with HS. (**A**) Pediatric curve. (**B**) Adult group curve.

**Figure 8 biomedicines-12-01607-f008:**
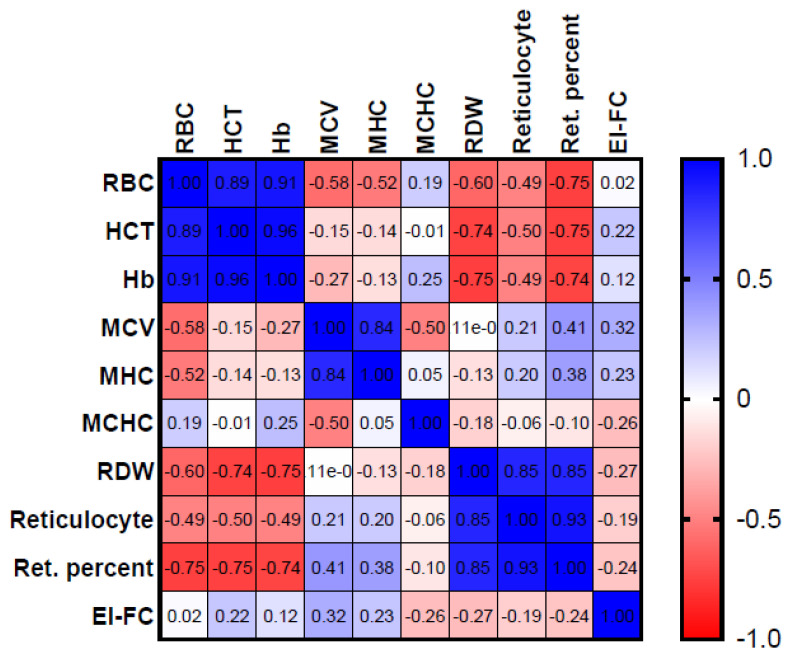
Correlation between EI-FC (EI-FC HS patients) and hematological parameters.

**Table 1 biomedicines-12-01607-t001:** RBC indices of HS patients categorized based on whether they had undergone splenectomy before performing the osmotic gradient.

RBC Indices	Non-Splenectomized (*n* = 10)			Splenectomized (*n* = 4)		
	Mean	Maxime	Minim	Mean	Maxime	Minim
RBC (×10^6^/µL)	3.7	4.8	2.5	4.8	5.8	3.6
HCT (%)	31.2	38.3	23.1	41.3	46.7	38.6
Hb (g/L)	11.1	13.0	7.6	14.6	16.9	13.2
MCV (fL)	85.0	101.2	74.1	88.8	108.7	73.8
MHC (pg)	30.3	36.0	27.0	31.5	37.7	26.9
MCHC (g/dL)	35.8	38.5	29.9	35.6	36.5	34.7
RDW (%)	18.7	21.5	14.3	13.3	14.8	12.5
Reticulocyte (×10^9^/µL)	351.8	533.4	205.6	123.4	173.4	73.5
Ret (%)	9.7	14.3	4.5	2.7	4.1	1.3

**Table 2 biomedicines-12-01607-t002:** Patient’s EMA dye test results.

Adult’s (*n* = 5)			Children (*n* = 9)
Patients	Percent Decrease Expression	Patients	Percent Decrease Expression
A-HS1	*	C-HS1	28.5
A-HS2	*	C-HS2	34.5
A-HS3	30	C-HS3	25
A-HS4	29.68	C-HS4	24
A-HS5	21	C-HS5	41.5
		C-HS6	33
		C-HS7	26.3
		C-HS8	23
		C-HS9	30

Note: * These patients were diagnosed in their childhood by the clinical feature and osmotic fragility test.

**Table 3 biomedicines-12-01607-t003:** Ratio EI-FC in patients with spherocytosis hereditary.

EI-FC at m308Osm/kg	Control	Spherocytosis Patients	Ratio
Children	0.127	0.08	0.62
Adults	0.126	0.10	0.79

## Data Availability

The original raw data and materials presented in the study can be made available upon request. Further inquiries can be directed to the corresponding author.

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
