# Peer review of "Flow Cytometry as a New Accessible Method to Evaluate Diagnostic Osmotic Changes in Patients with Red Blood Cell Membrane Defects"

_biomedicines, 2024, doi:10.3390/biomedicines12071607_

Round 1

Reviewer 1 Report

Comments and Suggestions for Authors

The manuscript “Flow Cytometry as a New Accessible Method to Evaluate 1 Diagnostic Osmotic Changes in Patients with Red Blood Cell Membrane Defects” presents a method that is easy to implement and has high sensitivity and specificity that could complement the study of hereditary spherocytosis. It is an interesting study, and the manuscript may be accepted for publication after minor revisions. Followings are some suggestions for further revisions.

1. The abstract can be improved. Using it consists of background, method and content, results and significance of the study.

2. Please cite some related references for some explanations, for example, the first paragraph of introduction section.

3. The key innovations of present work can be emphasized in the last paragraph of introduction section.

4. There are too many paragraphs for each section, please combine some of them.

5. The quality of figures can be improved. For example, the resolution of Figure 2, the first graphic of Figure 5A can be adjusted,

6. Please check and unify the units. For example, the “ml” and “mg/ml” can be changed to “mL” and “mg/mL”, respectively.

7. It is suggested to add a separated conclusion section, and the shortcomings of present work can be emphasized in the conclusion section.

8. Please check and revise some minor problems, for example, “Membranopathies, are heterogeneous disorders.” (Line 11); “were able to” (Line 15) can be changed to “can”; “Were determinate for……” (Line 171); the “n” of “n=10” and “n=4” should be in Italics.

9. Please avoid using first-person narratives for a scientific paper. It is better not to start a sentence with numeral number, such as Line 134.

10. The “Supplementary Materials:” section can be deleted if there were not any supplementary materials.

Comments on the Quality of English Language

1. Please check and revise some minor problems, for example, “Membranopathies, are heterogeneous disorders.” (Line 11); “were able to” (Line 15) can be changed to “can”; “Were determinate for……” (Line 171); the “n” of “n=10” and “n=4” should be in Italics.

2. Please avoid using first-person narratives for a scientific paper. It is better not to start a sentence with numeral number, such as Line 134.

Author Response

For research article

Response to Reviewer 1 Comments

1. Summary

2. Questions for General Evaluation

Reviewer’s Evaluation

Response and Revisions

Does the introduction provide sufficient background and include all relevant references?

Can be improved

In the introduction, a reference to the first paragraph has been added, and in the final paragraph, an attempt has been made to integrate a main idea as suggested by the reviewer.

Are all the cited references relevant to the research?

Can be improved

We have been added

Is the research design appropriate?

Yes

Are the methods adequately described?

Yes

Are the results clearly presented?

Yes

Are the conclusions supported by the results?

Yes

3. Point-by-point response to Comments and Suggestions for Authors

Comments 1: The abstract can be improved. Using it consists of background, method and content, results and significance of the study.

Response 1: The summary has been improved reduced and make more comprehensible . We agree with this comment. – page number 1 line 12-26

Comments 2:   Please cite some related references for some explanations, for example, the first paragraph of introduction section.

Response 2: Agree. In the first paragraph, a bibliographic reference has been added, about the characteristics of the red blood cell and the importance of its plasma membrane. – page number 1 line 33-34

Comments 3: The key innovations of present work can be emphasized in the last paragraph of introduction section.

Response 3: We have added a final paragraph in the introduction that reaffirms the importance of the work. We agree with this comment. – page number 2, lines 70-73

Comments 4: There are too many paragraphs for each section, please combine some of them.

Response 4: We have reduced the text and improved the clarity of our results. We agree.

4o

Comment 5:  The quality of figures can be improved. For example, the resolution of Figure 2, the first graphic of Figure 5A can be adjusted,

Response 5: We improved the quality of figure 2 and figure 5, in the case of figure 5, we adjusted the size accordingly the format page.

Comment 6:     Please check and unify the units. For example, the “ml” and “mg/ml” can be changed to “mL” and “mg/mL”, respectively.

Response 6: Agree. We have been checking all units and unify.

 Comment 7: It is suggested to add a separated conclusion section, and the shortcomings of present work can be emphasized in the conclusion section.

Response 7: Agree. In the final part we have adding a conclusion section for explanation as a meaning of the article. – page number 12, lines 358-361

Comment 8: Please check and revise some minor problems, for example, “Membranopathies, are heterogeneous disorders.” (Line 11); “were able to” (Line 15) can be changed to “can”; “Were determinate for……” (Line 171); the “n” of “n=10” and “n=4” should be in Italics.

Response 8: Agree. These changes have been done. page number 1, line 11; 15, page 4 lines 175-176.

Comments 9:  Please avoid using first-person narratives for a scientific paper. It is better not to start a sentence with numeral number, such as Line 134.

Response 9: Agree. The have changed the start a sentence. Page 4 line 136.

Comment 10: The “Supplementary Materials:” section can be deleted if there were not any supplementary materials.

Response 10: Agree. The have eliminated that section.

4. Response to Comments on the Quality of English Language

Point 1:

Response 1:    The article has a lot paragraphs with confuse sentence, I so grateful for your comments and I will try to improve the grammar of the text.

5. Additional clarifications

Reviewer 2 Report

Comments and Suggestions for Authors

In this manuscript by Beltran et al, the authors have described their use of flow cytometry to assess changes in RBCs using light scatter patterns as the cells are exposed to different osmolarities flanking the iso-osmolar point of 308 mOsm/Kg. By assessing shape changes, the authors aim to identify osmolar levels that reveal reproducible changes in the elongation index ratios (EI, or EI-FC for flow cytometry), calculated from the major vs. minor axes of the ellipse that forms in RBCs as determined by forward vs. side scatter, respectively. Their results begin by testing healthy RBC samples from adults vs. children, each assessed at a range of osmolarities flanking the iso-osmolar point (i.e., hypo-osmolar vs. hyper osmolar), which revealed interesting curves for each that could be used to indicate a normal distribution pattern. The authors then compared mean responses at each osmolarity, with the aim of identifying differences to the iso-osmolar point of 308 that are reproducible. The data indicated that several levels were significantly different at both hypo- and hyper-osmolar conditions, specifically 176 vs. 458 mOsm/Kg points (although others appeared statistically significant). The authors next tested HS patient samples, although did not actually show the range of osmolarities as shown with the healthy controls. Nonetheless, there were statistically different levels at 176, 308 (the iso-osmotic point), and 458 mOsm/Kg, but correlations to the EMA test suggests the results from children samples were more specific. Finally, comparisons to other hematologic parameters failed to find a correlation to their designed EI-FC test. Overall, the test for elongation indexes does appear to provide another parameter to test RBCs when diagnosing a membranopathy such as hereditary spherocytosis, but there are issues that need to be addressed. As detailed below (but this is not comprehensive), there are multiple areas of the text that must be edited for grammatical errors or to fix confusing areas. More importantly, distributions of EI-FC levels at the osmolarities shown for healthy controls is important so show for the patient samples (adult vs. child), as this might add to the knowledge of how the RBCs in patients are influenced by the different osmotic pressures. Furthermore, how do the data presented here compare to the gold standard of an OGE test? Perhaps a direct comparison would add confidence to their results showing differences of EI ratios at 308, 176 or 458 mOsm/Kg, and whether such differences are predictive of a disease state. Other issues are listed below.

Lines 34-36, the sentence is grammatically incorrect as written, and how is the conjunctival complex different from the ankyrin complex? This should be clearly described.

Lines 40-41, the semi-colon should be removed after "hydrophobic bonds", and the end of this sentence requires another quotation after "spectrin repeats".

Lines 42-43, the sentence is a bit confusing as written - how is the "open dimer" created with respect to the tetramer, and what final structure contains the "balance" between the two structures, which provides flexibility to the structure?

Line 44, remove the semi-colon after "RBC membranopathies".

Line 45, suggest adding "of RBCs" after "alter the cytoskeletal structure....", plus add "of these cells" after "transport function". Or change the sentence to make it clear as to how disorders in the structure or other functions lead to RBC membranopathies.

Line 52, the first sentence should reiterate the specific disorder here; is this referring to the general "RBC membranopathies", or HS in particular? This should be clearly stated in this new paragraph, and it should be reiterated in subsequent paragraphs (those starting with "Diagnosis" or "the diagnosis", as examples) for clarity.

Line 82, the authors need to restate the model system being analyzed - reiterate that RBCs will be analyzed by flow cytometry for the identified changes, comparing HS patients to healthy controls.

Figure 1, is the X-axis labeled correctly? Frequence?

Line 171, the first sentence needs editing - starting with "Were determinant" is grammatically incorrect, and the next sentence shows "Table 1." standing alone - perhaps place between parentheses.

Line 180-181, the first sentence of this paragraph is grammatically flawed and difficult to follow, so must be edited. 

Results, Section 3.2, where is the data for the EMA tests on patient vs. healthy controls? If a thorough study for using flow cytometry on RBCs from HS patients, the actual data from each patient should be shown.

Line 197, the sentence is grammatically incorrect, perhaps change to "ratio of these two parameters will allow for the calculation of the elongation index....". 

Line 202-204, Figure 2 legend, the second sentence requires editing as it’s grammatically incorrect and confusing. Also, what are the values in the Y-axis for the graph - this is important to show as it will describe the axis shown in the healthy samples in Fig 3.

Figure 3 legend, line 214, why is the figure referring to patients with HS? The data and text indicate that the curves are from either healthy adults (A) or healthy children (C).

Section 3.3.1, third paragraph (referring to Fig. 4), the following would be helpful. First, set the stage for the figures - are they simply showing the same EI ratio curves in A vs. B (or C vs. D), with just statistical comparisons between the iso-osmolar point of 308? Why show these comparisons? The importance of showing these comparisons should be stated.

Figure 5 legend, there are multiple issues. First, define the terms used in each figure for "CTLA" vs. "HSA", and CTLC and HSC (assume control vs. HS patient, adult vs. child, respectively, but must define). Second, the second sentence is very confusing and needs to be edited. Third, the fourth sentence is very confusing and should be reworded to concisely indicate the use of the ROC curves (and relate this to that described in the text).

Lines 260-263, the text is very confusing as to what differences are being described. The previous sections described differences in the EI ratios between the iso-osmolar level (308) and the hypo- or hyper-osmolar levels (e.g., 176 or 458, respectively), but now are they referring to these differences, or differences between each EI ratio between healthy vs. patient samples, each at a particular osmolarity, not by comparison to the iso-osmolar levels? A better description that carefully states the comparisons being used would be helpful.

Figure 5, why did the authors not show a representative curve from each type of patient, e.g., adult vs. child? This would be helpful to show prior to, or even within the same figure, to show how the patient samples compare to controls at all of the tested osmolarities, as is shown in Fig 4 for the healthy controls. 

Lines 270-271, the authors now start comparing the EI ratios to the EMA tests, but the rationale here is confusing. This is also why the actual EMA test results would be helpful, so that the reader can understand the logic of the comparison.

Lines 278-280, again without the EMA results, the significance of this sentence is lost.

Under Discussion, as stated above, how did their flow cytometry data from FSC/SSC parameters under different osmolarities compared to the gold-standard test for RBCs of OGE? Also, how dependable is their observed differences at 308 mOsm/Kg? Finally, without a comparison of unknown samples (e.g., a blind test) that compares results from healthy samples vs. HS samples, it is difficult to predict the accuracy of this test in a clinical setting. 

Also under Discussion, the final statement that the presented technique "can assess accurately the functional capacity of RBCs" (lines 378-379) is a bit misleading - while the shape of cells is revealed and assessed at the different osmolarities, or at the iso-osmolar point of 308 mOsm/Kg, how this relates to actual functional aspects of RBCs is unclear. Does the loss of an elliptical shape at different osmolarities actually reveal a function of RBCs? Finally, can the authors use imaging flow to basically show the same thing, but with individual pictures to determine correlations between the EI ratios used here vs. actual RBC shape? Such results might add to the impact of their study; at least some comparisons to such studies would be good to discuss.

Comments on the Quality of English Language

As stated in comments to the authors, there are multiple grammatical errors and confusing sentences, so a careful edit for English is warranted.

Author Response

For research article

Response to Reviewer 2 Comments

1. Summary

Thank you very much for taking the time to review this manuscript. Please find the detailed responses below and the corresponding revisions/corrections highlighted/in track changes in the re-submitted files. I express my gratitude for the comprehensive summary of the article and the insightful comments provided on the text. Before submitting the manuscript to the journal, I had similar considerations, and therefore, I am grateful for the time you have dedication.

2. Questions for General Evaluation

Reviewer’s Evaluation

Response and Revisions

Does the introduction provide sufficient background and include all relevant references?

Can be improved

[Please give your response if necessary. Or you can also give your corresponding response in the point-by-point response letter. The same as below]

Are all the cited references relevant to the research?

Yes

Is the research design appropriate?

Can be improved

Are the methods adequately described?

Yes

Are the results clearly presented?

Can be improved

Are the conclusions supported by the results?

Can be improved

3. Point-by-point response to Comments and Suggestions for Authors

Comments 1: Lines 34-36, the sentence is grammatically incorrect as written, and how is the conjunctival complex different from the ankyrin complex? This should be clearly described.

Response 1: Thank you for pointing this out. We have rewritten the introduction to enhance the clarity of the article, focusing solely on hereditary spherocytosis without discussing the vertical and horizontal interactions that maintain the structure and function of the membrane.

page number 1

Comments 2: Lines 40-41, the semi-colon should be removed after "hydrophobic bonds", and the end of this sentence requires another quotation after "spectrin repeats".

Response 2: Thank you for pointing this out. We have rewritten the introduction to enhance the clarity of the article, focusing solely on hereditary spherocytosis without discussing the vertical and horizontal interactions that maintain the structure and function of the membrane.

. Page number 1

Comments 3: Lines 42-43, the sentence is a bit confusing as written - how is the "open dimer" created with respect to the tetramer, and what final structure contains the "balance" between the two structures, which provides flexibility to the structure?

Response 3. Thank you for pointing this out. We have rewritten the introduction to enhance the clarity of the article, focusing solely on hereditary spherocytosis without discussing the vertical and horizontal interactions that maintain the structure and function of the membrane.

Page number 1 33-34

Comments 4: Line 45, suggest adding "of RBCs" after "alter the cytoskeletal structure....", plus add "of these cells" after "transport function". Or change the sentence to make it clear as to how disorders in the structure or other functions lead to RBC membranopathies.

Response 4: We have modified the sentence to better explain the nature of membrane disorders, which are associated with structural changes or alterations in the function of certain membrane transporters. Page number 1, paragraph 3, and lines 35-39.

Comments 5: Line 52, the first sentence should reiterate the specific disorder here; is this referring to the general "RBC membranopathies", or HS in particular? This should be clearly stated in this new paragraph, and it should be reiterated in subsequent paragraphs (those starting with "Diagnosis" or "the diagnosis", as examples) for clarity.

Response 5:  To separate the terms Membranopathies from Hereditary Spherocytosis, we have divided them into two paragraphs. In paragraph 6, we exclusively discuss hereditary spherocytosis Page number 1, paragraph 3, and lines 40-45.

Comments 6: Line 82, the authors need to restate the model system being analyzed - reiterate that RBCs will be analyzed by flow cytometry for the identified changes, comparing HS patients to healthy controls.

Response 6: We confirm the type of technique to be used, the measurement to be conducted, and the population on which it will be implemented. Page number 2, paragraph 7, and lines 70-73.

Comments 7: Figure 1, is the X-axis labeled correctly? Frequence?

Response 7: In Figure 1, X represents the frequency of cases included in the study. Page number 3.

Comments 8: Line 171, the first sentence needs editing - starting with "Were determinant" is grammatically incorrect, and the next sentence shows "Table 1." standing alone - perhaps place between parentheses.

Response 8: We have corrected the grammatical error and revised the paragraph accordingly. Additionally, we have adjusted the citation style for Table 1. Page number 5, and lines 169-171.

Comments 9: Line 180-181, the first sentence of this paragraph is grammatically flawed and difficult to follow, so must be edited.

Response 9: We have rewritten this paragraph.  Page number 5, lines 171-177

Comments 10: Results, Section 3.2, where is the data for the EMA tests on patient vs. healthy controls? If a thorough study for using flow cytometry on RBCs from HS patients, the actual data from each patient should be shown.

Response 10: A table with the results of the patients has been added.   Page number 5, Table 2.

Comments 11: Line 197, the sentence is grammatically incorrect, perhaps change to "ratio of these two parameters will allow for the calculation of the elongation index....".

Response 11 The figure citation has been rewritten for clarity and simplicity, and the phrase has been omitted as the calculation of the ratio is explained in the results. Page number 6, Lines 192-197.

Comments 12: Line 202-204, Figure 2 legend, the second sentence requires editing as it’s grammatically incorrect and confusing. Also, what are the values in the Y-axis for the graph - this is important to show as it will describe the axis shown in the healthy samples in Fig 3.

Response 12:   The paragraph has been rewritten to provide a clearer explanation of the technique's principle. Pages number 6-7, Lines 199-205.

Comments 13: Figure 3 legend, line 214, why is the figure referring to patients with HS? The data and text indicate that the curves are from either healthy adults (A) or healthy children (C).

Response 13:   It's an error; the figure represents the curve of the control group, and it has been modified. Page number 7 Lines 217-221.

Comments 14: Section 3.3.1, third paragraph (referring to Fig. 4), the following would be helpful. First, set the stage for the figures - are they simply showing the same EI ratio curves in A vs. B (or C vs. D), with just statistical comparisons between the iso-osmolar point of 308? Why show these comparisons? The importance of showing these comparisons should be stated.

Response 14:  Because the iso-osmolar point refers to the physiological condition that enables us to measure variations between different osmolarities, and the differences between the iso-osmolar point and the various osmolarities are significant, it would allow us to compare with other points. However, it would also be necessary to observe the variability of these points according to the distribution and how they behave within the ROC curve.

Comments 15: Figure 5 legend, there are multiple issues. First, define the terms used in each figure for "CTLA" vs. "HSA", and CTLC and HSC (assume control vs. HS patient, adult vs. child, respectively, but must define). Second, the second sentence is very confusing and needs to be edited. Third, the fourth sentence is very confusing and should be reworded to concisely indicate the use of the ROC curves (and relate this to that described in the text).

Response 15:   The terms of each figure and their corresponding meanings have been clarified. Secondly, the paragraph discusses the non-parametric test used to compare the means of the control group with the patient group at each of the 3 selected osmolarities, and we have not changed it. In the third and fourth paragraphs, we refer to the ROC curve for the iso-osmolar point between the groups and the use of the EMA test as a point of comparison. Pages number 9-10 Lines 261-273.

Comments 16: Lines 260-263, the text is very confusing as to what differences are being described. The previous sections described differences in the EI ratios between the iso-osmolar level (308) and the hypo- or hyper-osmolar levels (e.g., 176 or 458, respectively), but now are they referring to these differences, or differences between each EI ratio between healthy vs. patient samples, each at a particular osmolarity, not by comparison to the iso-osmolar levels? A better description that carefully states the comparisons being used would be helpful.

Response 16. We have rewritten the results of Figure 5, adding the ROC curve for osmolarities 176 and 458 mOsm/kg and presenting the results in a clearer manner.

Comments 17: Figure 5, why did the authors not show a representative curve from each type of patient, e.g., adult vs. child? This would be helpful to show prior to, or even within the same figure, to show how the patient samples compare to controls at all of the tested osmolarities, as is shown in Fig 4 for the healthy controls.

Response 17:   Se ha incluido una figura donde destaca la media del grupo de control sano en adultos y niños y la curva de cada paciente estudiado Page number 11 Lines 287-294.

Comments 18: Lines 270-271, the authors now start comparing the EI ratios to the EMA tests, but the rationale here is confusing. This is also why the actual EMA test results would be helpful, so that the reader can understand the logic of the comparison.

Response 18:   The comparison is made with the results of the EMA binding test, which have been added in Table 2. Using the criterion of an EMA equal to or above 21 as positive and the EI-FC ratio.

4. Response to Comments on the Quality of English Language

Point 1:

Response 1:    (in red)

5. Additional clarifications

[Here, mention any other clarifications you would like to provide to the journal editor/reviewer.]

In this section, I would like to clarify the comments made by reviewer 2 to enhance the understanding of the direction or focus presented in this work. While the work does contain a series of grammatical errors that may hinder comprehension of the article, we apologize for this and have worked to rectify these errors to ensure quality in the text.

An important point raised by reviewer 2 is the need to display the results obtained from the calculated EI-FC values, particularly regarding the distribution of these values. Therefore, we have added a supplementary page where tables displaying the normality distribution, assessed using the Kolmogrov and Smirnof test for both the control and patient groups, are presented. At the 176 mOsm/Kg point, we observed a division of the population into two cellular populations, one with greater size/complexity and others with less.

While reviewer 2 rightly suggests that the optimal method for comparison would be through an osmotic gradient by ectacytometry, this comparison was not feasible. However, our comparison was conducted using a method known for its high specificity and sensitivity, making it the method of choice in erythropathology laboratories dealing with spherocytosis cases, alongside molecular diagnosis. Notably, the latest cases added to this study involved 2 pediatric patients with suspected spherocytosis, and a pregnant woman with a family history. In these cases, the diagnostic algorithm was applied for anemia study alongside the presented technique. All three patients exhibited an EI-FC lower than the mean of the control group and a loss of expression of EMA-5 equal to or above 21%.

Although we performed a ROC curve with the ratios obtained at points 176 and 458 mOsm/Kg, a decrease in the sensitivity and specificity of the technique was observed. Hence, only the comparison with the iso-osmolar curve has been depicted. If the reviewer suggests, we could include it as well.

Reviewer 3 Report

Comments and Suggestions for Authors

Dear Authors

I carefully read and checked the article. Title is good, abstract is enough. Introduction is well written. Although the number of patients included in the study seems to be small, it can be considered sufficient for such cytometric studies. The methodology is proper, the statistical methods used are correct, the cytometric tests are appropriate and successful. The findings are clearly expressed. I believe that the tables and figures are successful. The discussion is very detailed and good comparisons are made. With this kind of cytometric study, we have learned that flow cytometry can be used to detect osmotic changes in erythrocyte membrane defects, which is an important contribution to the literature.

Author Response

For research article

Response to Reviewer 3 Comments

1. Summary

2. Questions for General Evaluation

Reviewer’s Evaluation

Response and Revisions

Does the introduction provide sufficient background and include all relevant references?

Yes

Are all the cited references relevant to the research?

Yes

Is the research design appropriate?

Yes

Are the methods adequately described?

Yes

Are the results clearly presented?

Yes

Are the conclusions supported by the results?

Yes

Note from reviewer

Dear Authors I carefully read and checked the article. Title is good, abstract is enough. Introduction is well written. Although the number of patients included in the study seems to be small, it can be considered sufficient for such cytometric studies. The methodology is proper, the statistical methods used are correct, the cytometric tests are appropriate and successful. The findings are clearly expressed. I believe that the tables and figures are successful. The discussion is very detailed and good comparisons are made. With this kind of cytometric study, we have learned that flow cytometry can be used to detect osmotic changes in erythrocyte membrane defects, which is an important contribution to the literature.

Response

As authors, we appreciate your time and dedication. You are correct regarding the number of patients for this study; although it is small, we observed a trend and significant differences between controls and patients. This allows us to suggest that this technique could be helpful in the diagnosis of Hereditary Spherocytosis due to equipment availability and cost-effectiveness.

Round 2

Reviewer 2 Report

Comments and Suggestions for Authors

The authors have made significant improvements to the manuscript and addressed the concerns of this reviewer, however there are some minor issues that should be addressed as follows:

Line 98, "noteed" is misspelled.

Line 276, capitalize T in "table 2".

Lines 280-294, Table 2, the authors need to add beneath the table what the asterisks are indicating for patients A-HS1 and A-HS2.

Line 305, "RBC" should be plural, and "there" should be "them", as its referring to the RBCs, e.g.,. "positions of the RBCs causing them to assume an elliptical shape...."

Lines 309-310, the sentence might be easier to read as "To measure cytometer changes of RBCs in a hypo-osmolar, hyper-osmolar, and iso-ismolar media,....".

Lines 312-314, "will allows for the calculation", the "allows" should be singular. Also, why did the authors remove "index" with the EI-FC, as this indicates Elongation index-flow cytometry? Suggest adding back the "index".

Lines 364-388, and Figure 4, the edited text is appreciated, but for clarity the authors might direct the reader to each side of the curves in Figure 4, referring to the hypo-osmolar points as the "left bars" on Figure 4A and 4B, while the hyper-osmolar points can be indicated as "right bars" for each curve. This will help guide the reader to the statistical data shown in Fig. 4.

Lines 444-447, Figure 5 legend, the authors need to indicate the two sets of graphs for "A" and "B", by simply indicating that those under "A" indicate adult (A), and those under "B" indicate children (C).

Figures 6 and 7 and associated text, the reviewer appreciates the inclusion of ROC curves for each comparisons (hyper-, iso-, and hypo-osmolarities, Fig. 6), and means showing direct comparisons of the curves for patient vs. control groups (Fig. 7). These certainly add confidence to the data.

Line 632, this reviewer still finds the phrase "precisely assess the functional capacity of red blood cells" to be misleading, in that the assay simply assesses the morphologies of RBCs in healthy vs. patient samples, which only reveals whether the cells exhibit abnormal morphologies under the stresses of flow cytometry. This test provides no measure of oxygen delivery, passage through capillaries, deformability, or any other functional aspect of RBCs, and therefore the authors cannot state that the assay provides a functional measurement without data to show that there is a correlation to actual "function".

Note throughout: The authors first define “RBC” as red blood cell, which is correct, but then redefine this or sometimes use it interchangeably with “red blood cells” (e.g., plural), vs. RBCs. Be sure to review the document to make sure that these are used correctly, and that there are not repetitive explanations of RBCs, as done in the Discussion (see lines 524, 537, 539, and 542 as examples).

Comments on the Quality of English Language

English is fine, just a few minor typographical errors that have been noted.

Author Response

Response to Reviewer 2 Comments

1. Summary

2. Questions for General Evaluation

Reviewer’s Evaluation

Response and Revisions

Does the introduction provide sufficient background and include all relevant references?

Yes

Are all the cited references relevant to the research?

Yes

Is the research design appropriate?

Yes

Are the methods adequately described?

Yes

Are the results clearly presented?

Yes

Are the conclusions supported by the results?

Yes

3. Point-by-point response to Comments and Suggestions for Authors

Comments 1: [Line 98, "noteed" is misspelled..].

Response 1: Thank you for pointing this out. I/We agree with this comment. It has been corrected.

Comments 2: Line 276, capitalize T in "table 2"

Response 2: Agree. It has been corrected.

Comments 3. Lines 280-294, Table 2, the authors need to add beneath the table what the asterisks are indicating for patients A-HS1 and A-HS2.

Response 3: Agree, To enhance understanding of the table, a note has been added indicating that these two patients were diagnosed bases on clinical features and the osmotic fragility test.

Comments 4: Line 305, "RBC" should be plural, and "there" should be "them", as its referring to the RBCs, e.g.,. "positions of the RBCs causing them to assume an elliptical shape...."

Response 4: We have corrected this grammatical error; RBCs have been written in the plural form, and "there" has been changed to "them."

Comments 5: Lines 309-310, the sentence might be easier to read as "To measure cytometer changes of RBCs in a hypo-osmolar, hyper-osmolar, and iso-ismolar media,....".

Response 5: We agre, we have changed the sentence as you suggested to make it more understandable.

Comments 6: Lines 312-314, "will allows for the calculation", the "allows" should be singular. Also, why did the authors remove "index" with the EI-FC, as this indicates Elongation index-flow cytometry? Suggest adding back the "index".

Response 6: We agree. The term “allows” has been changed to the singular forma, and the word “index” has been added to indicate the “elongation index in cytometry”.

Comments 7: Lines 364-388, and Figure 4, the edited text is appreciated, but for clarity the authors might direct the reader to each side of the curves in Figure 4, referring to the hypo-osmolar points as the "left bars" on Figure 4A and 4B, while the hyper-osmolar points can be indicated as "right bars" for each curve. This will help guide the reader to the statistical data shown in Fig. 4.

Response 7: We agree. For better understanding of the figure, the right bars indicating significant differences in the hyperosmolar curve between adult and pediatric populations have been specified, as well as the left bars indicating significant differences in the hypo-osmolar curve.

Comments 8: Lines 444-447, Figure 5 legend, the authors need to indicate the two sets of graphs for "A" and "B", by simply indicating that those under "A" indicate adult (A), and those under "B" indicate children (C).

Response 8: We have indicated the letter “A” for the adult population and the letter “B” for the pediatric population.

Comments 9: Figures 6 and 7 and associated text, the reviewer appreciates the inclusion of ROC curves for each comparison (hyper-, iso-, and hypo-osmolarities, Fig. 6), and means showing direct comparisons of the curves for patient vs. control groups (Fig. 7). These certainly add confidence to the data.

Response 9: We appreciate this comment from reviewer2.

Comments 10: Line 632, this reviewer still finds the phrase "precisely assess the functional capacity of red blood cells" to be misleading, in that the assay simply assesses the morphologies of RBCs in healthy vs. patient samples, which only reveals whether the cells exhibit abnormal morphologies under the stresses of flow cytometry. This test provides no measure of oxygen delivery, passage through capillaries, deformability, or any other functional aspect of RBCs, and therefore the authors cannot state that the assay provides a functional measurement without data to show that there is a correlation to actual "function".

Response 10: The reviewer is correct in this comment. This test does not measure a function of the red blood cell, but rather changes in its morphology as it passes through a laser beam in the cytometer. For this reason, we have changed the sentence to a term that better defines the test “The new test allows us to demonstrate differences between normal and abnormal RBCs populations and permits sensitive and specific discrimination, making it useful for the diagnosis of HS”.

Comments 11:  The authors first define “RBC” as red blood cell, which is correct, but then redefine this or sometimes use it interchangeably with “red blood cells” (e.g., plural), vs. RBCs. Be sure to review the document to make sure that these are used correctly, and that there are not repetitive explanations of RBCs, as done in the Discussion (see lines 524, 537, 539, and 542 as examples).

Response 11. The reviewer is correct, and we have corrected the grammatical errors and the repetition of “ red blood cells”, and included the acronym.

4. Response to Comments on the Quality of English Language

Point 1:

Response 1:  They were corrected.

5. Additional clarifications
